# Cyp33 binds AU-rich RNA motifs via an extended interface that competitively disrupts the gene repressive Cyp33-MLL1 interaction *in vitro*

**Neil R. Lloyd[¤], Deborah S. Wuttke** *

Department of Biochemistry, UCB 596, University of Colorado Boulder, Boulder, Colorado, United States of America

¤ Current address: KBI Biopharma, Louisville, CO, United States of America
* Deborah.wuttke@colorado.edu

## Abstract

Cyp33 is an essential human cyclophilin prolyl isomerase that plays myriad roles in splicing and chromatin remodeling. In addition to a canonical cyclophilin (Cyp) domain, Cyp33 contains an RNA-recognition motif (RRM) domain, and RNA-binding triggers proline isomerase activity. One prominent role for Cyp33 is through a direct interaction with the mixed lineage leukemia protein 1 (MLL1, also known as KMT2A) complex, which is a histone methyltransferase that serves as a global regulator of human transcription. MLL activity is regulated by Cyp33, which isomerizes a key proline in the linker between the PHD3 and Bromo domains of MLL1, acting as a switch between gene activation and repression. The direct interaction between MLL1 and Cyp33 is critical, as deletion of the MLL1-PHD3 domain responsible for this interaction results in oncogenesis. The Cyp33 RRM is central to these activities, as it binds both the PHD3 domain and RNA. To better understand how RNA binding drives the action of Cyp33, we performed RNA-SELEX against full-length Cyp33 accompanied by deep sequencing. We have identified an enriched Cyp33 binding motif (AAUAAUAA) broadly represented in the cellular RNA pool as well as tightly binding RNA aptamers with affinities comparable and competitive with the Cyp33 MLL1-PHD3 interaction. RNA binding extends beyond the canonical RRM domain, but not to the Cyp domain, suggesting an indirect mechanism of interaction. NMR chemical shift mapping confirms an overlapping, but not identical, interface on Cyp33 for RNA and PHD3 binding. This finding suggests RNA can disrupt the gene repressive Cyp33-MLL1 complex providing another layer of regulation for chromatin remodeling by MLL1.

## Introduction

Proline isomerases are a superfamily of chaperones that, by catalyzing the *cis* to *trans* isomerization of the ubiquitous amino acid proline, regulate the folding of proteins that then influence a wide range of biological pathways [1, 2]. First discovered as the target of the human

---

---

**Data Availability Statement:** SELEX data are available on Zenodo (DOI: 10.5281/zenodo. 4450582). NMR data are available on Figshare (DOI: 10.6084/m9.figshare.13607741.v1).

**Funding:** DSW, MCB 1716425, The National Science Foundation, https://www.nsf.gov NL, T32 GM-065103, The National Institutes of Health, General Medicine Division, https://www.nih.gov.

**Competing interests:** The authors have declared that no competing interests exist.

immunosuppressive therapeutic cyclosporin [3], there are 17 known human cyclophilins with overlapping but not completely redundant functions [4]. Variation in the surface residues near the active sites and the presence of accessory domains within many cyclophilins tailor the action of each cyclophilin to specific targets [4]. Cyp33 is a nuclear localized proline isomerase, and is unusual in possessing an N-terminal RNA-recognition motif (RRM) in addition to its C-terminal cyclophilin domain [5–8]. Cyp33 is an essential gene in humans [9, 10], broadly conserved in *Drosophila melanogaster* and *Caenorhabditis elegans*, and highly conserved in mammals with a wide range of implicated functions in both splicing [11–13] and chromatin remodeling [6–8].

Cyp33 is best known for its central role in regulating the activity of the myeloid/lymphoid or mixed lineage leukemia (MLL1) complex [6–8, 14]. MLL1 is an epigenetic transcriptional co-activator/co-repressor and H3K4 methyltransferase [15, 16] involved in epigenetic maintenance, negatively regulating transcription of critical genes such as *HOX9A* and *MEIS1* [7, 15, 17]. The regulation of MLL1 by Cyp33 depends on both the RRM and cyclophilin domains of Cyp33 [8]. The RRM domain of Cyp33 specifically interacts with the third plant homeodomain (PHD3) finger of MLL1, contributing to Cyp33 recruitment to and association with the MLL1 complex. The isomerase activity of the cyclophilin domain is involved in rearrangement of MLL1 which exposes the RRM binding interface to allow stable association of Cyp33 with MLL1 [6–8, 14]. Moreover, deletion of the isomerase domain or inhibition by cyclosporin A disrupts this repressive activity of Cyp33 *in vivo* [14]. Persistent binding of Cyp33 to MLL1 likely introduces conformational changes within the complex and other associated binding partners leading to Cyp33-isomerase-dependent gene regulation such as that mediated by histone deacetylase 1 (HDAC1) [14] and BMAL1 [14, 18–20]. Notably, deletion of the PHD3 domain in MLL1 results in leukemia and cell line immortalization [21–23], and insertion of PHD3 even within oncogenic MLL1 protein fusions restores Cyp33 mediated repression of genes involved in cancer cell immortalization [23]. Thus, Cyp33 is a key regulator of epigenetic silencing at prominent developmental genes whose activity is disrupted in human leukemic cancers.

The ability of Cyp33 to interact with RNA introduces an additional layer of regulation for Cyp33 activities well-suited to regulation of transcription. *In vitro*, Cyp33 binds AU-rich RNA sequences, including polyA, polyU, purified mRNA, and the polyadenylation sequence AAUAAA [7, 24]. Moreover, mRNA binding activates the isomerase activity of Cyp33, although the mechanism for this activation is unknown [24]. Consistent with a key role for RNA binding, Cyp33 has ubiquitously been identified as an RNA-binding factor in global RNA-proteome crosslinking studies [25, 26]. Examination of the RRM domain shows it contains the canonically conserved amino acid residues typical of RNA binding [27]. One proposed model for the role of RNA binding is to disrupt the PHD3-RRM interface and reset MLL1 to its native conformation [6]. However, only the affinity for the AAUAAA ligand has been measured, weakly interacting with a ~200 μM $K_D$ of unclear biological relevance. This RNA interaction is readily disrupted by the tighter binding PHD3 domain which binds in the low micromolar range [6, 7].

To better define the RNA-binding specificity and mechanism of RNA-binding activation of the enzymatic activity of Cyp33, we performed RNA-SELEX against the full-length Cyp33 protein. Systematically sampling protein and salt conditions in 6 parallel selections, we have identified several RNA sequences with increased affinity compared to the reported RNA ligand, including an aptamer sequence which binds with a ~7 μM $K_D$. This sequence has an extended binding interface compared to the previously characterized ligand, resulting in chemical shift changes in residues altered upon PHD3 binding. NMR characterization of this ligand demonstrates that RNA is capable of directly disrupting the Cyp33-MLL1 interaction and implicates

additional residues outside the RRM domain in contributing to RNA affinity, suggesting possible mechanisms for the observed activation of Cyp33 by RNA. These studies provide structural insights into the tuning of Cyp function through competitive RNA and peptide binding.

## Materials and methods

### Protein expression and purification

Full-length Cyp33 was cloned into pET30b with a 6x-His maltose binding protein (His-MBP-FL-Cyp33) tag and into pET28b with a 10x-His Small Ubiquitin-like Modifier tag (His-SUMO-FL-Cyp33). In addition, 6x-His-FL-Cyp33 and 6x-His-Cyp33-RRM (both in pET28b with a thrombin-cleavable His tag) were generously gifted by the Kutateladze lab [7] for use in the SELEX and NMR experiments. The cyclophilin domain was cloned into pET15b with a C-terminal 6x-His tag (Cyp33-Cyp-His). PHD3 was cloned into pET28b with a 10x-His SUMO tag.

Plasmids (~50 ng) were transformed into BL21 (DE3) *E. coli* and selected on LB plates supplemented with kanamycin (for all full-length Cyp33 constructs, Cyp33-RRM, and SUMO-MLL1-PHD3) or ampicillin (for Cyp33-Cyp). Single colonies were then picked for a 40 mL 37˚C overnight growth with the same antibiotic selection. Using 10 mL of the overnight growths, 1L growths were inoculated and grown in 2L baffled flasks containing the respective antibiotic at 37˚C and shaken at 180 rpm for 2–3 hrs to an O.D.$_{600}$ of 0.6–0.8 before being induced with 1 mM IPTG. After induction, the growth temperature was decreased to 18–20˚C and cells were grown for 18-20hrs before being harvested by centrifugation and frozen at -20˚C.

Frozen pellets were thawed in lysis buffer (100 mM Tris pH 8 at 4˚C, 1 M NaCl, 10% glycerol, 0.1% Triton-X100, 10 mM imidazole; 40–50 mL final volume) supplemented with a Roche EDTA-free protease inhibitor tablet before lysis by sonication. Lysed cells were then spun at 15K RPM and the supernatant fraction was incubated with Ni-NTA beads equilibrated with lysis buffer for 0.5–1 hr. After 3 washes with 50 mL lysis buffer, the captured protein fraction was eluted with lysis buffer supplemented with 350 mM imidazole in two 15–20 mL fractions. Eluted protein was then concentrated down to ~1.5–2 mL in Sartorius concentrators (10K MWCO for FL-Cyp33 and Cyp33-Cyp constructs and 5K MWCO for Cyp33-RRM). For PHD3 and FL-Cyp33 used for binding, the SUMO-tag was cleaved off through the addition of His-tagged Ulp1 during the concentration step and dialyzed overnight back into the lysis buffer prior to an additional flow through a Ni-NTA column. The concentrated protein was then further purified with size-exclusion chromatography on an Akta FPLC through injection onto a Superdex G75 (Cyp33-RRM, MLL1-PHD3 and 6x-His-FL-Cyp33) or Superdex G200 column (His-MBP-FL-Cyp33 and His-SUMO-FL-Cyp33) (both columns from GE Healthcare). After elution, fractions were combined and concentrated to ~400 μM to 2 mM, with yields typically between 6 to 18 mg/L, and flash frozen in liquid nitrogen and stored at -70˚C for later use. For isotopically labeled protein, protein was grown in minimal media with ($^{15}NH_4)_2SO_4$ or $^{15}NH_4Cl$ as the sole nitrogen source [28] and otherwise expressed and purified as described above.

### SELEX experiments

The following steps (except amplification of the initial library) were repeated 15 times. For the first 8 rounds, selections were alternatingly performed against 10x-His-SUMO-FL-Cyp33 (odd rounds) and 6x-His-MBP-FL-Cyp33 (even rounds) to eliminate RNAs that bound the tag. For rounds 9–15, selections were performed against 6x-His-thrombin-FL-Cyp33. Primer and library sequences are shown in **S1 Table**.

**PCR amplification of initial library.** The initial randomized DNA template of the library (IDT) was produced by PCR amplification of the complementary DNA. 2.0 nmols of the DNA template was used to generate the initial library with a random region of 50 nucleotides, flanked by two constant regions with sequences expected to have low propensity for forming secondary structures. This template was split into ten 100 μL aliquots of the following reaction conditions: 2 μM DNA template, 5 μM primers, 1 mM dNTPs, 1X Taq Buffer (10 mM Tris pH 8.3, 50 mM KCl, 1.5 mM MgCl$_2$), and 1U/50 μL Taq polymerase. Following a 5 min incubation at 95°C, PCR amplification was performed for 10 cycles of 95°C for 45s, 55°C for 45s, 68°C for 45s.

**RNA transcription.** RNA was *in vitro* transcribed using T7 RNA polymerase. PCR template (10% of final volume) was added to a reaction volume with the final buffer of 40 mM Tris pH 7.9, 24 mM MgCl$_2$, 1 mM DTT, 2 mM spermidine. 1U/100 μL of T7 RNA polymerase and inorganic pyrophosphatase were then added and incubated at 37°C for 4–16 hours.

**RNA purification.** RNA was purified by gel purification. RNA was mixed with 2X loading buffer (95% formamide, 0.5M EDTA, 0.1% bromophenol blue) and loaded onto an 8M urea, 8% polyacrylamide denaturing slab gel and run at 20-30W for 2–4 hours. RNA bands were visualized by UV shadowing on a Fluor-Coated TLC Plate (Fisher Scientific), cut out, and then crushed and soaked 2 hours to overnight in 0.5X TE pH 7.5 buffer. The gel particles were filtered using 0.22 μM cellulose-acetate filters (ThermoScientific), before being concentrated in a 5K MWCO centrifuge concentrator (Sartorius). Once the RNA volume reached ~0.5 mL, 1 mL IDT nuclease free water was added and spun again, with the process repeated three times to remove residual urea. RNA concentration was assessed at A$_{260}$ using a NanoDrop Spectrometer using extinction coefficients predicted by IDT Oligo Analyzer.

**Constant region primer annealing and pre-selection against Co-NTA beads.** RNA was first refolded by incubation at 80°C for 5 minutes followed by snap cooling on ice in the presence of 2X molar ratio of DNA primers complementary to both constant regions. This RNA, at a concentration of 7.7 μM for the first round of selection and 1.1 μM for all subsequent rounds, was then pre-incubated with Co-NTA beads (Jena Biosciences) in 1.1X selection buffer for 15 minutes. The Co-NTA beads were then separated to the side of the tube with a magnetic stand while the supernatant was added to the binding equilibrium reaction.

**Binding equilibrium, washing, and elution for selection.** Protein at several concentrations (100 nM, 500 nM, or 1000 nM) was incubated in 1X SELEX buffer (conditions in **S2 Table**) for 1 hour with ~7 μM pre-selected RNA for the first two rounds and ~1 μM pre-selected RNA for subsequent rounds. Co-NTA beads were then added and incubated for 15 minutes prior to separation with a magnetic stand. Supernatant was removed and the Co-NTA resin was washed 3X with wash buffer. After the final wash, 20 μL of 1X SELEX buffer supplemented with 350 mM imidazole was then used to resuspend the Co-NTA resin. After 15 minutes, the resin was again separated with a magnetic stand and the supernatant used as the input for a reverse transcriptase reaction.

**Reverse transcription (RT)-PCR.** First 1 μM RT primer complementary to the 3' region of the RNA was added to 14 μL of eluted RNA. In a thermocycler, the protein was denatured at 80°C for 10 minutes and then cooled to 4°C over ~15 minutes. After annealing, 4 μL of 5X RT buffer (100 mM Tris pH 7.5, 50 mM NaCl, 50 mM MgCl$_2$, 5 mM DTT) was added along with 1 μL of 10 mM dNTPs and 1U of reverse transcriptase. The RT reaction was performed at 60°C for 20 minutes followed by 80°C for 10 minutes utilizing a thermostable group II intron reverse transcriptase [29]. The full RT reaction was then used as the template for a 500 μL PCR reaction in 100 μL aliquots with the following reaction conditions: 20μL/500μL RT-PCR template, 1 μM primers, 0.5 mM dNTPs, 1X Taq Buffer (10 mM Tris pH 8.3, 50 mM KCl, 1.5 mM

MgCl$_2$), and 1U/50 μL Taq polymerase. PCR amplification was performed for 10 cycles of 95˚C for 45s, 55˚C for 45s, 68˚C for 45s.

**Preparation of SELEX libraries for sequencing.** To submit our SELEX libraries to high-throughput sequencing, Illumina adapter sequences had to first be appended onto the sequences for proper adherence to the Illumina cell. We accomplished this by PCR amplification of our libraries with primers containing the Illumina adapter sequences and sequences complementary to the constant regions. In both cases, the 5' P5 Illumina adapter with a T7-5' constant region sequence required only 1 step of PCR for addition while the 3' P3 Illumina adapters also containing 12mer indexing barcodes required 2 PCR-steps for addition. Details of the sequences submitted to sequencing are highlighted in **S1 Table**.

**PCR step 1.** Using the P5'-T7-5' constant primer and our 3' adapter primer, we amplified our libraries with 8 cycles of 95˚C for 45s, 55˚C for 45s, 68˚C for 45s with 100 μL reaction volumes of the following concentrations: 1 μM input library, 5 μM primers, 1 mM dNTPs, 1X Taq Buffer (10 mM Tris pH 8.3, 50 mM KCl, 1.5 mM MgCl$_2$), and 1U/50 μL Taq. The PCR products were then cleaned up using a E.Z.N.A. Cycle Pure Kit (Omega).

**PCR step 2.** We used the product from PCR Step 1 as the template for the second step to add the P3-barcode indexing primer. Using the P5'-T7-5' constant primer and our P3-barcoding primer, we amplified our libraries for 8 cycles as in step 1. The PCR products were then cleaned up as in step 1.

**Pooling and quality control.** The resulting libraries were then quantified using a Nanodrop spectrometer and pooled together at roughly equimolar concentrations. The combined pool was then gel purified to select the correctly sized products on a native 1X TBE 8% polyacrylamide gel. Following a crush and soak in 1X TE pH 7, the pooled sample was filtered using a 0.22 μM cellulose-acetate filter (ThermoScientific) and submitted to the CU Boulder BioFrontiers Sequencing Facility for quality control and sequencing. The size distribution of the pool was quantified using a High Sensitivity D1000 ScreenTape system and the concentration was determined using Qubit Fluorometric Quantitation.

## Electromobility shift assays (EMSAs)

To quantify the binding affinity of the target proteins for RNA ligands, EMSAs were performed using radiolabeled RNA ligands produced by T7 polymerase *in vitro* transcription and purified protein. The 5' phosphate of transcribed RNA ligands was removed using calf intestinal phosphatase (CIP, NEB) and then replaced with $^{32}$P using T4 polynucleotide kinase (PNK, NEB) and $^{32}$P γ-ATP (Perkin Elmer). Labeled ligand with a final concentration of 5 nM was added to 2-fold serial dilutions of the purified protein ranging from 400 μM to 0 nM final concentration in SELEX buffer supplemented with 10% glycerol. Samples were loaded onto a 0.5X TBE 8% polyacrylamide gel while running at 200V at room temperature for 15–20 minutes. The gels were then dried and exposed on a phosphor screen and imaged on an Amersham Typhoon Imaging System. The resulting images were quantified in ImageQuant 5.0 and fit to the quadratic binding equation in Excel using Solver by minimizing the sum of the least squares difference between the data and fit. A minimum of 3 replicates were analyzed.

## NMR $^1$H-$^{15}$N HSQC experiments

All NMR experiments were performed at 25˚C on an Agilent DD2 600 MHZ spectrometer using a z-axis gradient HCN room-temperature probe. Data acquisition times ranged from 1.5 hr (Cyp33-RRM titrations with the selected RNA sequence SO-1, see **Table 1** below) to 16 hours (initial and final titration points). Varian BioPack pulse sequences were used with minor modifications. Protein concentrations typically varied from between 200 μM to 100 μM

**Table 1. Summary of $K_{D, Apparent}$ for SELEX oligonucleotides and round 0 library to FL-Cyp33.** SELEX Oligos (SO) correspond to the rank-sorted abundance of each sequence overall, *e.g.*, SO-1 is the most abundant sequence at the L500 condition.

| SELEX Oligo | $K_{D, Apparent}$ (μM) by EMSA | Sequence Length | Origin Condition |
|---|---|---|---|
| Initial RNA Pool | 50 ± 20 | 90 | N/A |
| SO-1 | 6.6 ± 0.6 | 50 | L500 |
| SO-3 | 32 ± 2.5 | 28 | multiple |
| SO-2 | 21 ± 3.1 | 50 | P500 |
| SO-4 | 55 ± 10 | 50 | L100 |
| SO-6 | 36 ± 10 | 50 | P1000 |
| SO-8 | 24 ± 13 | 51 | L500 |
| SO-9 | ~90 | 50 | P100 |

upon addition of ligand and performed in the SELEX buffer (**S2 Table**) supplemented with 10% $D_2O$. Spectra were processed using standard protocols with NMRPipe and analyzed in CcpNmr Analysis. Assignments for 55 of 81 non-proline residues of Cyp33-RRM were confidently transferred from Park et al. [6]. Unassigned peaks were due to ambiguity of overlapping chemical shifts of the native protein residues and additional peaks arising from the His-tag. The observed chemical shift change ($\Delta\delta_{obs}$) for each amino acid residue was calculated using the following equation [28] and mapped onto the NMR Cyp33-RRM structure (PDB: 2KYX) [6].

$$\Delta\delta_{obs} = \sqrt{\left(\Delta\delta_H^2 + \left(0.17 * \Delta\delta_N\right)^2\right)}$$

## NMR $^1$H-$^{15}$N HSQC competition experiments

The data collection and processing for $^1$H-$^{15}$N HSQC data for the competition experiment was as described for the titration experiments. Data on a 200 μM $^{15}$N-labeled Cyp33-RRM was first collected with a 1:1 molar ratio of SO-1. An equimolar concentration of MLL1-PHD3 was then added and data collected prior to the final addition of SO-1 to a final molar ratio of ~5:1 SO-1 to other components for data collection. Data collection times were 16 to 33 hrs.

## Results

### Parallel SELEX experiments against FL-Cyp33 at different conditions produced enrichment of AU-rich motifs

To systematically sample the selection space for Cyp33 RNA binding, we performed six parallel selections using a 50N randomized RNA library against Cyp33. Several strategies were employed to address known issues that bias selections. To determine if the sequences selected were dependent on the selection conditions, we varied both protein concentration (at 100 nM, 500 nM, and 1000 nM) and salt concentration (physiological salt; 135 mM KCl, 15 mM NaCl, and 1/3 physiological salt) to test the impact of overall stringency and electrostatics, respectively. The 50N random region was flanked by constant regions with a low propensity to form secondary structures (**S1 Table**). The constant regions were annealed to complementary DNA primers during the selection to mitigate interactions between the random and constant regions of the RNA ligands. To avoid potential His-tag occlusion by RNA binding, we attached the His-tags to larger solubility tags to create His-SUMO and His-MBP constructs, which spatially separated the affinity tag from the targeted binding interface. These His-SUMO and His-MBP constructs were alternatingly used as the SELEX protein target to prevent the enrichment of RNA aptamers for either solubility tag.

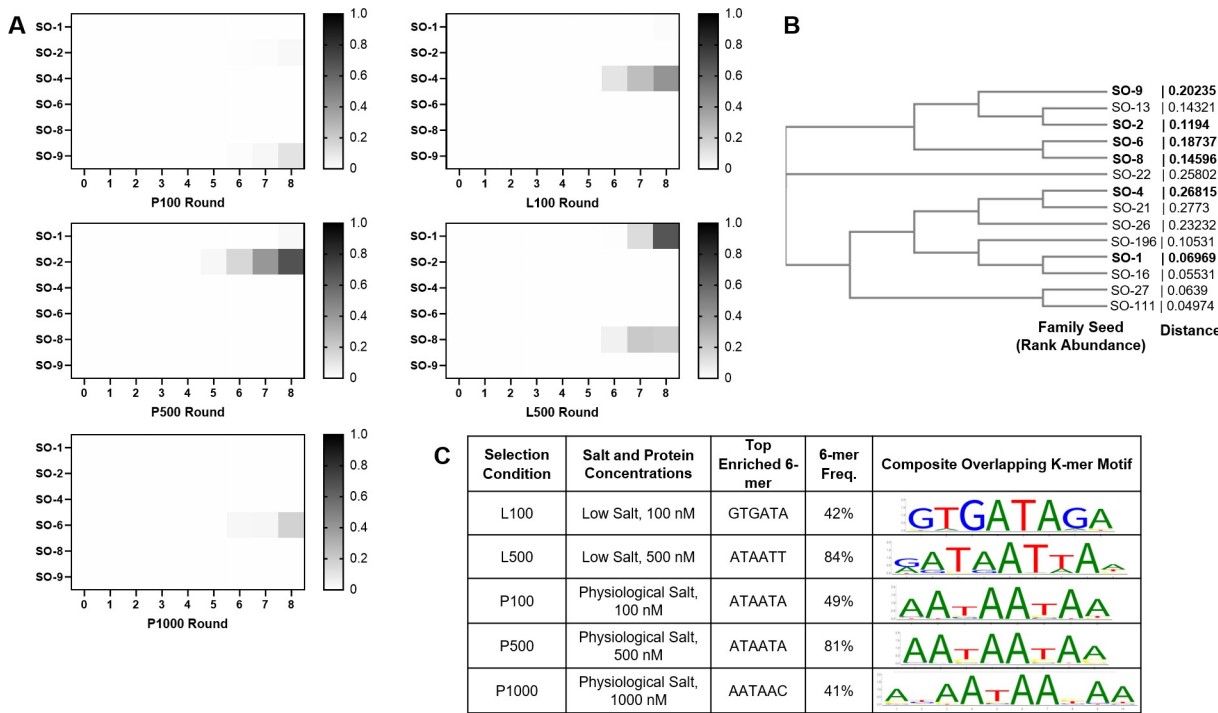

**Fig 1. Enrichment of selected RNA sequences. A)** Heatmaps of the fractional abundance of the top clusters of sequences for each selection condition (clusters as rows, rounds as columns). SELEX Oligo (SO) clusters are named based on the rank abundance of the seed cluster and shown as a fraction of the total pool (grayscale from 0 as white to 1 as black). Other are all other sequences that did not cluster within the top 14 clusters. **B)** Dendrogram of the SELEX Oligo seed sequences aligned by ClustalW [32] to illustrate their degree of conservation and inter-relatedness. The most abundant oligonucleotides (in bold) were validated by binding. **C)** AptaSUITE [33, 34] top enriched 6-mer sequences for each selection condition with the composite motif and frequency of overlapping k-mers shown as a motif logo. Low salt = L; Physiological salt = P; number corresponds to concentration of protein in nM for selection conditions L100, L500, L1000, P100, P500, and P1000. Selection conditions described in Materials and Methods.

Fifteen rounds of selection were conducted at all conditions using minor modifications of standard strategies as described in Materials and Methods. We leveraged the high read depth of deep sequencing to sequence all 90 SELEX round samples, as well as the initial library, using an Illumina NextSeq. From this sequencing we produced 184 M reads, 157 M of which passed quality filtering (Phred score >20, random read region > 10 nt), resulting in an average of 1.7 M usable reads per sample with a range of 1.2–3.8 M reads after demultiplexing and primer truncation. We then clustered all sequences from all conditions with total reads n > 5 for *de novo* clustering at an 80% similarity threshold using QIIME [30]. To identify the most abundant sequence clusters, we filtered for clusters comprising >0.05% of total sequences, revealing 14 highly abundant clusters. The most abundant sequence within each of these clusters were then used as the seed sequence for closed reference clustering of all sequences against those 14 sequence families [31]. This clustering revealed high over-selection of sequences and cross contamination between selection conditions after round 9 (and earlier for the low salt 1000 nM protein condition), thus data from round 9 on was excluded the subsequent analysis. Plotting the abundance of each family by condition and round up to round 8 (**Fig 1**) reveals that each condition (other than the excluded L1000 condition) produced at least one clearly enriched sequence cluster, suggesting that the range of conditions used in this selection for Cyp33 are appropriate for this system.

Since most RRMs that bind RNA do so at single-stranded regions of RNA and interact with 4 to 6 nucleotides through each RRM [27], we used AptaSUITE [33, 34] to determine whether there is any significant enrichment in particular 6-mer sequences through k-mer enrichment

analysis. Assessing enrichment in selected sequences over low abundance sequences (n < 10), we found significant enrichment of several overlapping 6-mers, revealing a preference for AU-rich sequences, often seen with several repeats amongst the highly abundant sequence reads (**Fig 1C**), and consistent with previous work that found Cyp33 preference for PolyA and PolyU RNA [24] as well as modest binding to the AAUAAA sequence [7]. Although the precise most prevalent k-mer sequence found varied somewhat (**Fig 1**), no significant variation in preferred sequence as a function of condition was observed.

## Top selected aptamer binds up to 30-fold tighter than the characterized AAUAAA ligand

To validate that the selected sequences bound FL-Cyp33, we *in vitro* transcribed the random region sequence of each of these aptamers and tested binding to FL-Cyp33. We first benchmarked the affinity of our enriched sequences by determining the affinity of FL-Cyp33 for the overall 90-nt RNA library pool, which binds with an estimated $K_D$ of 50 μM. Due to well-shifting behavior in the EMSA gels for the library pool, the stoichiometry of binding in these experiments is unclear and the fit $K_D$ value is likely the combination of many interactions of variable affinities. We then measured binding of the 50-nt selected regions from each of the top sequences of each selection condition, finding most meet or improve upon the mixture of the longer 90-nt initial library pool, with $K_D$s ranging from 6.6 to ~90 μM (**Table 1**). We note that the selection occurred in the context of 5' and 3' constant regions, such that it is possible several of these aptamers bind tighter within the secondary structural context of the constant region sequences. However, because of multiple conformations present within our RNA constructs and to simplify the RNA folding and help minimize the motifs involved in binding, these regions were not included during the binding validation.

Several conclusions can be drawn from these experiments. First, these data reveal that longer RNAs, which better represent cellular RNAs, interact with Cyp33 (**Fig 2A**) up to 30-fold more tightly than the 200 μM binding of the AAAUAA 6-mer previously reported [7]. While the improvements in affinity over the initial pool are modest (up ~7-fold tighter), these results indicate Cyp33 exhibits some sequence and/or structural specificity and suggest none of the aptamer sequences produced are cryptically tight interactions due to selection artifacts. Most notably, several of these RNA sequences bind comfortably within the low μM affinity range reported for the Cyp33-PHD3 interaction [6, 7]–indicating that our selections successfully enriched for aptamers with biologically relevant affinities.

## Identification of a truncated aptamer of SO-1, SO-3

Even though rounds 9–15 were generally not utilized due to cross contamination, so information from individual selection conditions cannot be obtained, analysis of the pooled sequences revealed a new sequence that was highly enriched. As this sequence is the 3rd most abundant sequence overall it is named SO-3. The 28-nt SO-3 is a truncated version of the most abundant and tightest binding sequence of the "winning" aptamer, SO-1, truncated suggesting that this sequence was a natural minimization of the binding site contained with SO-1. Corroborating this, the later rounds of selection show a high degree of nucleotide conservation within the 5' half of the selected aptamer region within SO-1 as opposed to more divergence at the 3' end (**Fig 2B**). Likewise, the sequencing depth provides similar data for SO-3 derived mutants, which shows nearly identical conservation of an AU-rich single-stranded region (**Fig 2B**). Binding of SO-3 to full-length Cyp33 reveals about a 5-fold decreased $K_D$ of 32 μM compared to SO-1 binding, but is notably tighter than other longer RNA constructs tested (**S2 Fig in S1 File**).

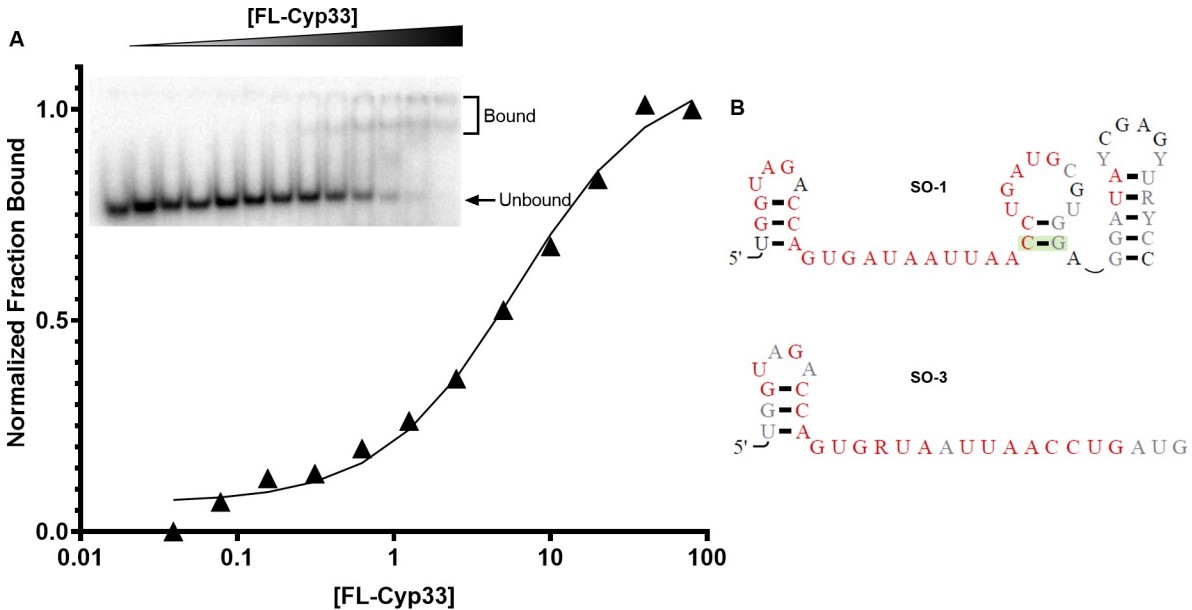

**Fig 2. SO-1 aptamer binds FL-Cyp33 with the tight affinity. A)** Representative EMSA data (inset) and fit of fraction bound for the quantification of $K_D$ for SO-1 binding. The leftmost lane of the gel is free RNA, while the rightmost lane is the highest protein concentration with each lane in between a 2-fold protein dilution of the lane to the right. Indicated bands quantified as free and bound SO-1 RNA. (Note, non-shifting RNA bands arising from RNA degradation are cropped out, the full gel is shown in **S1 Fig in S1 File**). Fit is to a standard binding isotherm as described in Materials and Methods. **B)** The predicted secondary structures of SO-1 and SO-3 selected RNAs with highly conserved base positions in red and poorly conserved positions in light gray. The green shaded base pair shows evidence of covariation. Secondary structures are predicated from phylogenetic correlations in R-scape [35] of MUSCLE-aligned [36] mutant sequences of SO-1 and SO-3.

## Secondary structure prediction suggests binding at single-stranded AU-rich sequences

Different selection conditions yielded different sequences with a range of binding affinities. To determine if they contained a conserved motif, we compared the lowest energy secondary structure predictions calculated with Vienna [31] for all of the SELEX oligos, shown in dot-bracket notation in **Table 2**.

Comparison of these predictions suggests that the AU-rich motifs of these sequences are conserved within single-stranded regions. We then tested whether this minimized single-strand region was necessary and sufficient for tight binding using synthesized oligos of the SO-1 singled-stranded AGUGAUAAUUAA and the AAUAAUAA k-mer commonly enriched among the other conditions (**Fig 1**). We were, however, unable to observe quantifiable binding with these ligands by gel shift. This is consistent with the previously characterized weak binding of 200 μM by AAUAAA, as the increased off-rate and reduced caging effect from the lower molecular weight ligand likely results in the bound complex dissociating faster than the timescale of the experiment. Thus, while a single-stranded AU motif is selected for, additional sequence/structure context are needed for high affinity binding. Given that SO-1 is both the most abundant selected sequence and shows the tightest binding, it is used in subsequent characterization.

## The Cyp33-RRM domain is not responsible for full RNA-binding activity

The selections were performed with the full-length Cyp33, which has both an RRM and a Cyp domain, thus the RNA could potentially interact with one or both domains. Binding by RNA

**Table 2. The predicted secondary structures and condition origins of the most abundant cluster seed sequences.**

| SELEX Oligo | Sequence–Lowest Energy Structure (kcal/mol) |
|---|---|
| | Dot-bracket Structure–Energy of prediction |
| SO-1 | UGGUAGACCAGUGAUAAUUAACCUGAUGCGUGGAGGAUAUCGAGUUGUCC -6.70 |
| | (((. . . .)))..((((((. . . .((((((. . . . .).))))))))))). -6.70 |
| | .((..(((. . .(((((. . . ..((. . .. .. .)). . . .)))))).))).. )) -6.40 |
| | (((. . .))). . .. . .. . ..((. . .. . .)).(((((. . . . .))))) -6.30 |
| SO-3 | UGGUAGACCAGUGAUAAUUAACCUGAUG -1.20 |
| | (((. . . .))). . .. . .. . .. . .. . . -1.20 |
| | . . .. . . .(((.(. . .. . .)))). . . -0.80 |
| SO-2 | CAGGUGUGUGACUACGAAAGAACAAUAAUAACACAAAAGAGUCCCGUGCC -5.40 |
| | ..(((((.(((((.(. . .. . .. . .. . .. . ..)))))).)))))) -5.40 |
| | ..(((.((.(((((.(. . .. . .. . .. . .. . ..)))))).)).))) -5.10 |
| SO-4 | UGGCCGGCCCAUCCCGACUGCCGGGUGAUAGACUCUUUAGCGAUUUAUGG -9.00 |
| | .(((((((. . . .))). . .)))((((. . ..)))). . .. . .. . .. . . -9.00 |
| | .(.(((((. . .. . .. . ..)))))).).((((((.((. . . .)))))))).. -9.00 |
| SO-6 | AGGUGCCUCAAAUCCGCAUAAGAAUAACAACAUGGAGUGAAGCGCUCCCC -10.50 |
| | .(((((.(((..(((((. . .. . .. . .. . ..)))).))).)))))). . . . -10.50 |
| | .(((((.(((..(((. . .. . .. . .. . .. ..))).))).)))))). . . . -10.40 |
| SO-8 | AGAACAAUAAUUACAAAGACUGAGCGUUUUAAAGUCUCCUCAUGUGCCCCC -3.90 |
| | . . .. . .. . .. .((((((. . .. . .. ..))))). . .. . .. . .. . . -3.90 |
| | . . .. . .. ..((((.(((((. . .. . .. ..))))). . . ..))). . .. . . -3.40 |
| SO-9 | AAAGUGAGAUAAGGUAACAACAAGAAUAAUAAUAUACCUAUCAUCUUGCC -8.50 |
| | . . .((((((((.(((((. . .. . .. .. . .. . ..))))). . .))))))). -8.50 |

The SO-# indicates the rank-sort abundance of each sequence. Representative secondary structures are shown for the lowest energy structures along with the kcal/mol energy of folding. RNA structure predictions are from Vienna [37].

is known to activate isomerase activity of the full-length protein [24]. Whether this isomerase activation is direct or indirect is not known. To determine which domain of the protein is responsible for RNA binding, we tested binding of the Cyp33-RRM alone to SO-1. To our surprise, the Cyp33-RRM binding to SO-1 was substantially reduced by ~30-fold to 180 +/- 15 µM compared to full-length Cyp33 binding (**S3 Fig in S1 File**). This weak binding is quite similar in affinity to the previous binding of the Cyp33-RRM reported with just AAUAAA (198 +/- 0.01 µM) [7] and suggests involvement of regions of full-length Cyp33 with the SO-1 ligand.

## The free Cyp33-RRM and Cyp33-Cyp subdomains behave independently in solution

As the interaction of the Cyp33-RRM alone with RNA is not sufficient to fully explain RNA binding, we employed NMR to characterize the complete FL-Cyp33 protein-RNA interaction. We used solution NMR to first understand the extent the two domains interact with each other in solution. Despite being 36 kDa, free FL-Cyp33 shows excellent signal-to-noise (**Fig 3A**). Comparison of the full-length $^{1}$H-$^{15}$N HSQC spectrum to that of the Cyp33-RRM (**Fig 3B**) and Cyp33-Cyp (**Fig 3C**) domains reveals that the sum of the two subdomain spectra largely recapitulate the FL-Cyp33 spectrum (Overlay in **Fig 3D**) with additional peaks from the Cyp33-Cyp domain likely arising from the C-terminal His-tag not shared by the FL-Cyp33. This indicates that the two subdomains behave largely independently of each other

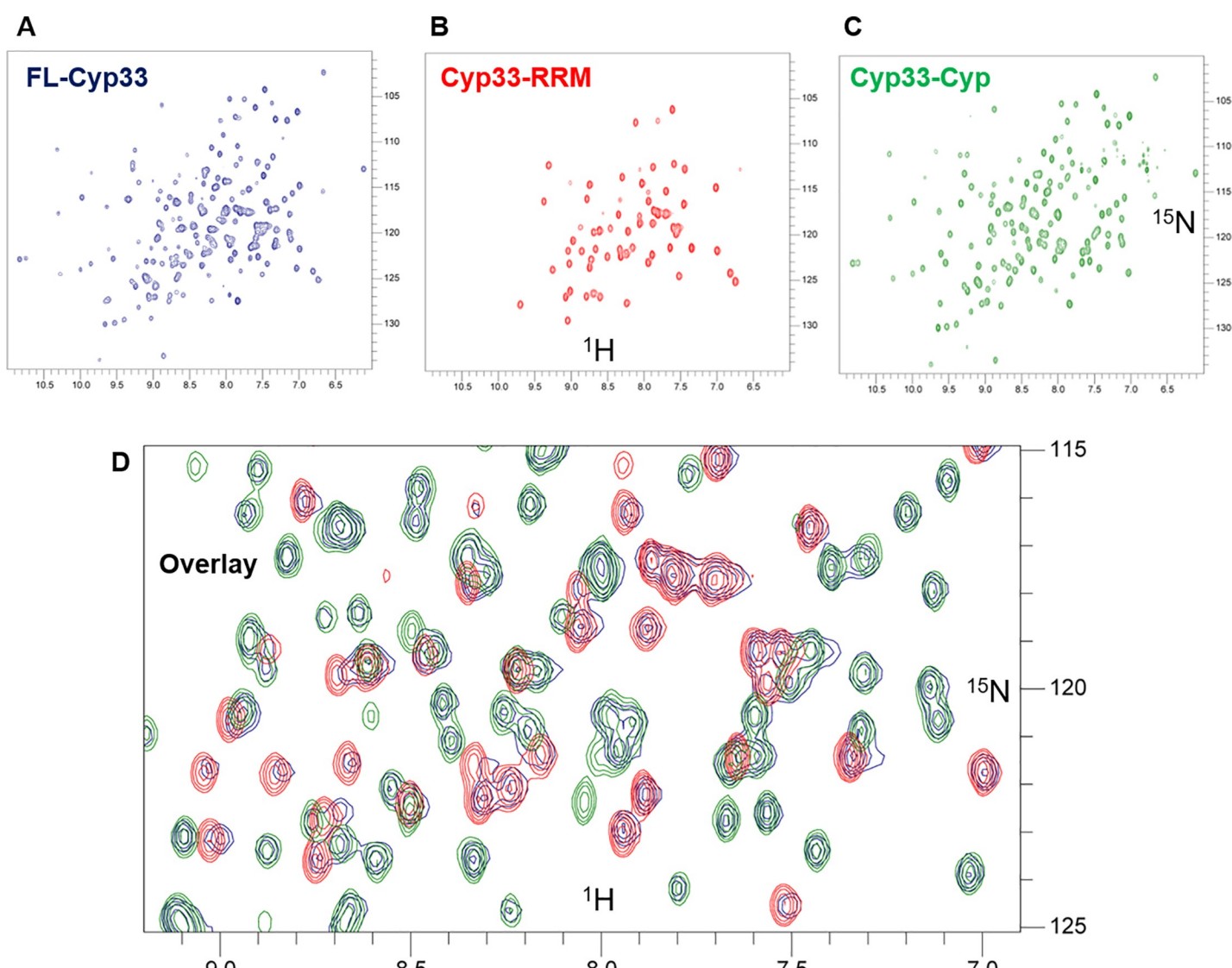

**Fig 3. Cyp33 subdomains behave independently of each other in solution. A)** $^1$H-$^{15}$N HSQC spectrum of FL-Cyp33 shown in blue. **B)** $^1$H-$^{15}$N-HSQC spectrum of Cyp33-RRM shown in red. **C)** $^1$H-$^{15}$N-HSQC spectrum of Cyp33-Cyp shown in green. **D)** Zoomed overlay of (A-C) showing that free FL-Cyp33 structurally behaves as the sum of the isolated subdomains (note that each blue peak overlaps with either a red or green peak). All proteins at 200 μM in SELEX buffer. A full overlay is shown in **S4 Fig in S1 File**.

in solution, consistent with the presence of a long ~50 residue linker between the two structured regions, which is largely not observed likely due to intermediate exchange properties. This subdomain independence suggests RNA must either interact outside the RRM domain or change the nature of the interaction between the two domains to explain the mechanism of isomerase activation by RNA [24].

## SO-1 does not interact with Cyp33-Cyclophilin domain in isolation

As our SELEX experiments were performed with FL-Cyp33, and RNA binding has been reported to activate isomerase activity by the Cyp domain, we tested whether RNA interacts with the Cyp33-Cyp domain alone using NMR titration. This strategy allows us to readily

detect the interaction even if it is quite weak. [15]N Cyp33-Cyp with SO-1 shows no significant perturbation of the [1]H-[15]N HSQC spectrum upon the addition of RNA (**S5 Fig in S1 File**), strongly suggesting that the RNA is not able to interact independently with Cyp33-Cyp at a biologically relevant affinity.

## Cyp33-RRM binds SO-1 aptamer with an extended interface compared to AAUAAA

The chemical shift changes observed upon addition of SO-1 to Cyp33 combined with the available structure [6–8] allow us to map the surface involved in RNA binding. Using the [1]H-[15]N chemical shift assignments made previously for Cyp33-RRM [7], we were able to reliably transfer ~67% of assignments for the native peaks we have observed (**S6 Fig in S1 File**). In mapping the significantly shifted residues onto the solved RRM structure [7] (**Fig 4A and 4B**), we observe that the changes upon SO-1 binding encompass the canonical RRM binding residues with additional chemical shifts in the loop between β-sheets 2 and 3. These changes could be due to either direct interaction or relayed conformational changes upon binding, we describe both of these as an extended binding surface. We compared these changes to the previously characterized binding surface of Cyp33-RRM with AAUAAA which is limited to the canonical residues and nearby residues (Tyr28, Phe68, Phe70, Asn99, K102, and Ile56) [7] and find that the region implicated in RNA binding is expanded in the context of binding to the longer SO-1 RNA. The SO-1 binding surface appears to encompass almost the entirety of the

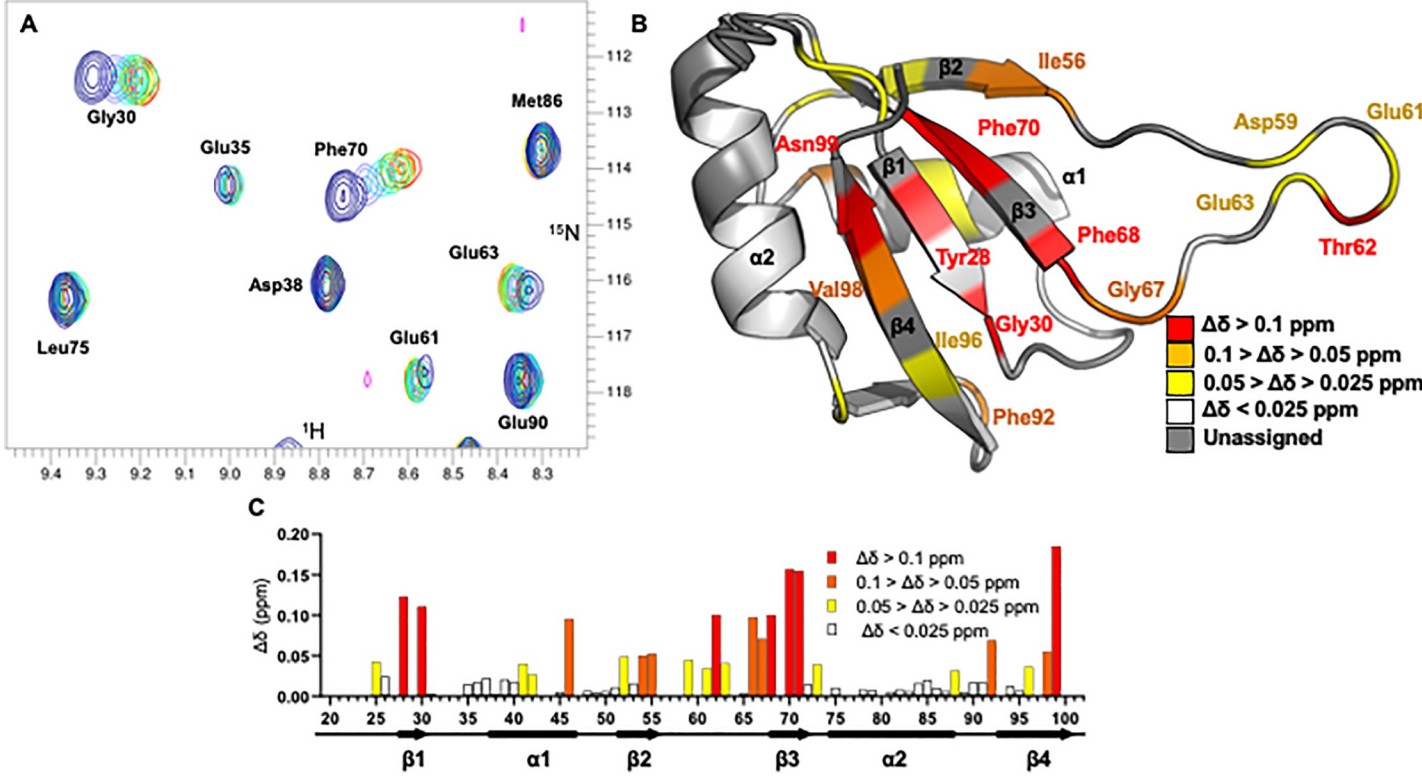

**Fig 4. [1]H-[15]N HSQC analysis and chemical shift mapping of Cyp33-RRM binding to SO-1. A)** [1]H-[15]N HSQC spectra of Cyp33-RRM with increasing molar ratios of SO-1; blue is free Cyp33-RRM, with indigo, cyan, green, yellow, orange, and red 0.25, 0.50, 0.75, 1.0, and 1.25 molar ratios of SO-1 to Cyp33-RRM, respectively. **B)** Assigned residues with significant chemical shift changes upon SO-1 binding is mapped onto the structure of Cyp33-RRM (2KYX) [6]. **C)** Histogram of chemical shift changes as a function of sequence, annotated with secondary structural elements.

binding face of the protein, with additional residue shifts in proximity of the canonical residues as well as in the loop between β-sheets 2 and 3. Notably, the extension of the SO-1 binding surface onto β-sheet 3 and the loop results in large overlap with the binding surface involved in the Cyp33-RRM and MLL1-PHD3 interaction [7]. As Cyp33 engages with longer RNAs *in vivo* [25, 26], we suggest this extended surface is more relevant for *in vivo* RNA binding.

## Selected RNAs interact beyond the RRM domain in FL-Cyp33

In addition to the extended RNA-binding features within the RRM, SO-1 binds FL-Cyp33 7-fold more tightly than Cyp33-RRM alone (see above), suggesting additional RNA-binding interactions beyond the Cyp33-RRM domain. To identify these, we characterized the $^1$H-$^{15}$N-HSQC spectrum of FL-Cyp33 bound to SO-1 and focused on chemical shift changes outside of the RRM region characterized above. Remarkably, addition of the 16 kDa SO-1 ligand to FL-Cyp33 results in resolvable resonance shifts, with many of the changes occurring in a fast-exchange regime (**S7 Fig in S1 File**). Comparison to the available NMR assignments for Cyp33-RRM [7] and the spectra of the individual subdomains (**Fig 3**) suggests the majority of the residues exhibiting significant changes upon binding map to the Cyp33-RRM domain (labeled) and shift similarly. Likewise, Cyp33-RRM resonances not impacted by SO-1 binding are likewise unaffected in the full-length protein complex, suggesting a similar binding mode. Notably, several additional resonances experience chemical shift changes upon binding (circled in pink) (**S7 Fig in S1 File**) suggesting further engagement of the full-length protein with the RNA, consistent with the enhanced binding affinity reported above.

## The Cyp33 RRM SO-1 binding interface overlaps with the MLL1-PHD3 binding interface and binds competitively

Comparison of the $^1$H-$^{15}$N HSQC spectra of Cyp33-RRM bound to MLL1-PHD3 or SO-1 reveals that, while many of residues in the overlapping interfaces both exhibit chemical shifts (**Fig 5A**), some of them are distinctly different. Mapping the SO-1 Cyp33-RRM interface onto the structure [6] alongside the known MLL1-PHD3 binding surface [7] (**Fig 5B**) highlights the extensive binding pocket overlap for the RNA and peptide ligands. Notably, Tyr28 is significantly shifted upon SO-1 binding but not MLL1-PHD3 binding (**Fig 5C**), allowing us to distinguish between populations of Cyp33-RRM bound to the different ligands. To determine if peptide and RNA binding are competitive, we titrated MLL1-PHD3 to an existing complex of Cyp33-RRM and SO-1. Rather than resulting in new chemical shifts indicative of a new complex or the loss of signal due to the size of a ternary complex, we observe split peak behavior (**Fig 5D**) indicative of a split population of Cyp33-RRM bound to either SO-1 or MLL1-PHD3. To confirm this result, we added excess of SO-1 to this sample, further pushing the equilibrium towards the Cyp33-RRM/SO-1 complex (**Fig 5E**), demonstrating that the two interactions are directly competitive even though they differ significantly in binding affinity (Cyp33-RRM binds the PH3 peptide with 1.9 μM affinity [7], significantly tighter than SO-1 at 180 μM, **S3 Fig in S1 File**). Thus, a large RNA can effectively disrupt the Cyp33-MLL1 interaction despite the weaker affinity of RNA for the RRM alone compared to the full-length Cyp33.

## Discussion

Cyp33 is an essential, two domain protein implicated in both splicing and transcriptional regulation [6–8, 11–14, 18]. The Cyp33-RRM domain binds short RNA and peptide elements competitively [6–8] and RNA-binding also activates the isomerase activity of the adjoining Cyp domain [24]. Comprehensive SELEX experiments reported here reveal that FL-Cyp33 binds a

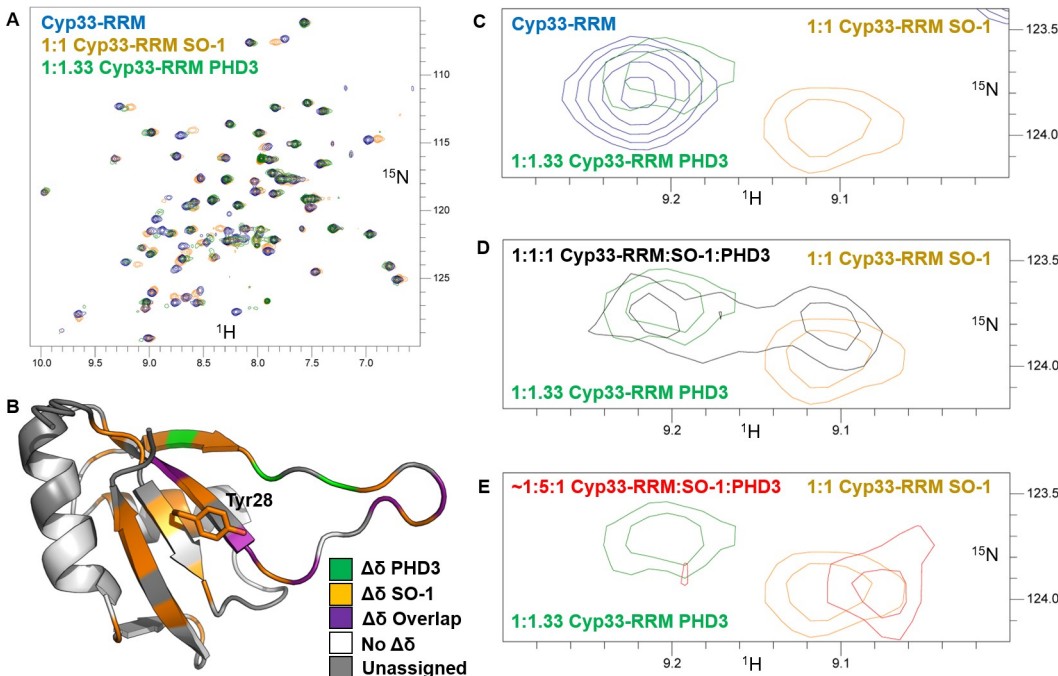

**Fig 5. $^1$H-$^{15}$N-HSQC of competitive binding of SO-1 and PHD3 to Cyp33-RRM. A)** Comparison of the $^1$H-$^{15}$N HSQC spectra of Cyp33-RRM free (blue) SO-1 bound (orange) and MLL-1-PHD3 bound (green). **B)** Comparison of the significant chemical shift changes upon SO-1 binding (orange), MLL-1-PHD3 (green) and overlapping residues (purple) mapped onto the Cyp33-RRM structure (2KYX) [6]. Residues without significant chemical shift changes are in white while unassigned residues are in gray. **C-E)** Zoomed overlay highlighting the $^1$H-$^{15}$N-HSQC chemical shift changes of Tyr28 upon ligand binding. **(C)** Free-Cyp33 (blue), MLL1-PHD3 bound (green), and SO-1 bound (orange). **(D)** MLL1-PHD3 bound (green), and SO-1 bound (orange), and both ligands 1:1:1 RRM:SO-1:PHD3 (black) **(E)** MLL1-PHD3 bound (green), and SO-1 bound (orange), and excess SO-1 added to the 1:1:1 sample in (D).

longer RNA more tightly than previously known, placing RNA and peptide binding in the same thermodynamic range. The core sequence enriched by the SELEX experiments is reminiscent of the AAUAA polyA sequence found in many mRNAs [38]. Cyp33 preference for this ubiquitous sequence suggests RNA binding by Cyp33 may be important in Cyp33 recruitment to the spliceosome and/or actively transcribed genes.

Comparison of the structural behavior of the Cyp33 subdomains relative to the full-length protein reveals that the RRM and isomerase domain are largely independent in solution. In conjunction with previous work showing the isomerase domain does not form stable, long-term interactions with MLL1 [8], this suggests the isomerase activity tethered to the MLL1 complex by the RRM domain is free to act upon other complex components within a relatively large distance range, constrained by the ~50 amino acid linker between the two Cyp33 domains. Cyp33 isomerase activity has been shown to regulate HDAC1 activity, effectively switching MLL1 from a general gene activator to a gene repressor [18].

Our data also address the mechanism of the activation of isomerase activity upon binding RNA [24]. Chemical shift changes in the FL Cyp33 spectrum upon binding SO-1 reveal that RNA-binding is not restricted to the RRM domain. Rather, additional interactions of the RNA with the linker and/or cyclophilin domain are consistent with our observation of tighter binding by the full-length protein for SO-1 compared to just the RRM domain. This work lays the foundation for identifying the residues that form the extended surface and design mutants that eliminate enzymatic activation upon RNA binding, as well as separation of function mutants distinguishing between RNA and MLL1 binding.

In many multi-domain proteins, RNA recognition by conserved RNA-binding domains is viewed as a separable activity, conferring RNA-recognition in a separable fashion. In the case of Cyp33, RNA-binding is instead integrated into the regulation of the activity of the protein. The Cyp33-RRM domain binds short RNA and peptide elements competitively [6–8] with overlapping but not identical protein surfaces (**Fig 5B**). The dual function of RNA and protein binding exhibited by the Cyp33-RRM is uncommon, but not unprecedented, among characterized RRMs [39]. For example, the Raver RRM1 is capable of binding both protein and RNA on separate faces of the RRM, although, unlike Cyp33, these interactions are likely not competitive [40, 41]. In a situation more similar to that observed for Cyp33, an RRM from the *Xenopus* polyA binding protein 2 (XePABP2) exhibits competitive binding between RNA and protein binding [42]. XePABP2 forms a homodimer, binding its own N-terminus rather than different protein as seen for Cyp33 with MLL1. RNA binding by XePABP2 disrupts this peptide interaction and thus the homodimerization [42], suggesting that competitive peptide and RNA binding are regulatory. How a similar type of competition between protein and RNA binding by Cyp33 regulates its function is not known but is consistent with its known activity at centers of RNA metabolism, such as transcription and splicing.

## Supporting information

**S1 File.**
(PDF)

**S1 Table. List of primers and oligos used in SELEX experiments.**
(DOCX)

**S2 Table. Selection conditions.**
(DOCX)

**S1 Raw images.**
(PDF)

## Acknowledgments

We would like to thank Robert Batey, Nicholas Parsonnet and Meagan Nakamoto for constructive feedback, Leslie Glustrom for comments on the manuscript and Nickolaus Lammer for assistance with manuscript revisions.

## Author Contributions

**Conceptualization:** Neil R. Lloyd, Deborah S. Wuttke.

**Data curation:** Neil R. Lloyd.

**Formal analysis:** Neil R. Lloyd, Deborah S. Wuttke.

**Funding acquisition:** Deborah S. Wuttke.

**Investigation:** Neil R. Lloyd, Deborah S. Wuttke.

**Methodology:** Neil R. Lloyd.

**Project administration:** Deborah S. Wuttke.

**Resources:** Deborah S. Wuttke.

**Supervision:** Deborah S. Wuttke.

**Visualization:** Neil R. Lloyd.

**Writing – original draft:** Neil R. Lloyd.

**Writing – review & editing:** Neil R. Lloyd, Deborah S. Wuttke.

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
