## [Decision Letter · Decision Letter 0]

4 Nov 2020

PONE-D-20-24207

Cyp33 Binds AU-Rich RNA Motifs via an Extended Interface that Competitively Disrupts the Gene Repressive Cyp33-MLL1 Interaction in vitro

PLOS ONE

Dear Dr. Wuttke,

Thank you for submitting your manuscript to PLOS ONE. After careful consideration, we feel that it has merit but does not fully meet PLOS ONE’s publication criteria as it currently stands. Therefore, we invite you to submit a revised version of the manuscript that addresses the points raised during the review process.

We apologize for the long time in handling the manuscript but it has proven really difficult to find suitable reviewers. The reviewer found the paper interesting, but there are a number of issues that need to be addressed before the manuscript becomes acceptable.

We look forward to receiving your revised manuscript.

Kind regards,

Oscar Millet

Academic Editor

PLOS ONE

Journal Requirements:

3.PLOS ONE now requires that authors provide the original uncropped and unadjusted images underlying all blot or gel results reported in a submission’s figures or Supporting Information files. This policy and the journal’s other requirements for blot/gel reporting and figure preparation are described in detail at https://journals.plos.org/plosone/s/figures#loc-blot-and-gel-reporting-requirements and https://journals.plos.org/plosone/s/figures#loc-preparing-figures-from-image-files. When you submit your revised manuscript, please ensure that your figures adhere fully to these guidelines and provide the original underlying images for all blot or gel data reported in your submission. See the following link for instructions on providing the original image data: https://journals.plos.org/plosone/s/figures#loc-original-images-for-blots-and-gels.

Reviewers' comments:

Reviewer's Responses to Questions

**Comments to the Author**

1. Is the manuscript technically sound, and do the data support the conclusions?

Reviewer #1: Partly

2. Has the statistical analysis been performed appropriately and rigorously? 

Reviewer #1: N/A

3. Have the authors made all data underlying the findings in their manuscript fully available?

Reviewer #1: No

4. Is the manuscript presented in an intelligible fashion and written in standard English?

Reviewer #1: Yes

5. Review Comments to the Author

Reviewer #1: Isomerase-dependent gene regulation is an interesting area Cyp33 is one such example. Cyp33 is peptidyl-prolyl isomerase best known for its central role in regulating the activity of the myeloid/lymphoid or mixed lineage leukemia (MLL1) complex. Cyp33 has two domains- a C-terminal the traditional cyclophilin domain that speeds up the isomerisation of a peptidyl prolyl bond, and an N-terminal RRM domain that binds short RNA sequences. Both the cyclophilin and RRM domains are required for Cyp33 regulation of MLL1. Cyp33 isomerises the peptidyl-prolyl bond in the linker region between the PHD3 and Bromo domain of MLL1. The RRM domains appears to have two roles: it interacts with the PHD3 domain as well as an RNA motif. In this paper, RNA-SELEX and deep sequencing was used to identify a strongly binding RNA motif (AAUAAUAA). NMR chemical shift mapping experiments using full-length Cyp33 gave a more complete picture of the RNA and PHD3 binding to Cyp33. The data showed that while the RNA sequence bound to the RRM domain, the residues involved are not wholly identical to those that bind to the PH3 domain of MLL1. Secondly, the RNA binds to residues outside the Cyp33 RRM domain. The results show the tuning of Cyp function through competitive RNA and peptide binding.

Line 248: what is SO-1? This was not defined.

Line 250: What is the physiological SELEX buffer – define this.

Line 252: Assignments for 55/81 non-proline: do you mean 55 out of 81? Clarify.

Line 442: it is surprising that there are very few resonances from the flexible 50 amino acid linker region. Is there a reason for this?

Figure 4: (a) Panels showing the NMR titration of the most significantly affect residues such as Y28, F68. F70, N99, etc should be shown. (b) The structure of the RRM domain should be annotated to show the numbering system for the �-sheets.

Line468: The chemical shifts changes should be displayed as a histogram, with the secondary structures of the RRM domain aligned with the residue numbers. This will an assessment of the significance of the chemical shifts outside the canonical RNA binding site of the RRM domain

Line 468: Using chemical shift changes to determine binding sites must be done judiciously. Shift changes caused by direct binding of ligand and indirectly due to relayed effects must be delineated, through analysing titration dependent changes. Small shift changes should be interpreted with caution. The best way to determined if the shift changes represent binding site is to perform site-directed mutagenesis.

Line 472: “Notably, the extension of the SO-1 binding surface onto β-sheet 3 and the loop results in large overlap with the binding surface involved in the Cyp33-RRM and MLL1-PHD3 interaction” – are the magnitudes of chemical shifts significant?

Line 485 and Supplementary Figure 7: The message here that extra regions of the FL Cyp33 protein is binding to the SO-1 is not discernible from this Figure. Panels showing the most significantly shift changes in the SO-1 titrations should be included. As the spectrum has not yet been fully assigned, concluding drawing form conclusions about expanded binding regions for SO-1 beyond the RRM domain is premature in the absence of addition data such as effects of mutagenesis. The fact that the linewidths of the spectra are similar means that binding is very weak indeed.

Line 495: “However, several chemical shifts not attributable to the known domains exhibit slow-exchange peak shifting behaviour. As these cannot be ascribed to either the Cyp33-RRM or Cyp33-Cyp domains, it suggests the involvement of the 46 amino acid linker in the interaction peaks”. How many peaks are in slow exchange? Do the binding site residues in the Cyp33 RRM domain also display slow exchange? If not, why are the linker residues in slow exchange? Slow exchange can be a result of sample conditions – was an excess of SO-1 added the Cyp33Fl sample to try and shift the equilibrium more in favour of the bound form?

Line502, Figure 5: The shift changes should be displayed as histograms for the results to be interpretable.

Line515: what are the relative binding constants between the Cyp33 RRM domain to SO-1 and to PHD3. Would you not expect the RRM domain to have more preference for the RNA than the PHD3? Was the reverse NMR competition experiment performed using an excess of PHD3 to compete out the RRM domain?

6. PLOS authors have the option to publish the peer review history of their article (what does this mean?). If published, this will include your full peer review and any attached files.

Reviewer #1: No

---

## [Author Response · Author response to Decision Letter 0]

21 Jan 2021

Response to Editor comments: 

All data are publicly available through the indicated DOIs.

A file named S1_raw_images has been uploaded that includes the requested raw gel images.

Response to Reviewer Comments:

We thank the reviewer for the careful consideration and review of this manuscript. We have addressed the points raised in review one-by-one as enumerated below. 

Overall, the review was favorable with some outstanding issues that we have addressed as described. Reviewers comments are italicized.

Line 248: what is SO-1? This was not defined. This was defined at a later point in the manuscript, and we have clarified the definition at Line248 as requested. 

Line 250: What is the physiological SELEX buffer – define this. This buffer is defined in Supplementary Table 2, we now specify this at Line250.

Line 252: Assignments for 55/81 non-proline: do you mean 55 out of 81? Clarify. Thank you for pointing out this ambiguity. We have clarified this statement as requested.

Line 442: it is surprising that there are very few resonances from the flexible 50 amino acid linker region. Is there a reason for this? This is not so uncommon in NMR, as the observation of flexible regions depends on both the exchange regime and pH of the solution. We have addressed this in the text, Line441.

Figure 4: (a) Panels showing the NMR titration of the most significantly affect residues such as Y28, F68. F70, N99, etc should be shown. (b) The structure of the RRM domain should be annotated to show the numbering system for the �-sheets. Regarding (a) we have included a superposition of the full spectrum of all titration points in Supplemental Figure 6. We believe this both clearly illustrates the chemical shift changes for all the significant shifters while also providing with all the data in an unbiased fashion. Regarding (b) we have made the requested changes to Figure 4, which we agree allow for clearer interpretation of the location of shifted sites on the canonical RRM domain. 

Line468: The chemical shifts changes should be displayed as a histogram, with the secondary structures of the RRM domain aligned with the residue numbers. This will an assessment of the significance of the chemical shifts outside the canonical RNA binding site of the RRM domain. We appreciate this suggestion and include the requested histogram as a new panel C of Figure 4.

Line 468: Using chemical shift changes to determine binding sites must be done judiciously. Shift changes caused by direct binding of ligand and indirectly due to relayed effects must be delineated, through analysing titration dependent changes. Small shift changes should be interpreted with caution. The best way to determined if the shift changes represent binding site is to perform site-directed mutagenesis. As pointed out by the reviewer, this is an important caveat, as the observed chemical shift changes could be due to indirect rather than direct interactions. We have clarified this point to include this alternative interpretation through new language at lines 469-472. While mutagenesis of all these residues could provide direct evidence of the binding interface, we encounter the same issue that mutants can have indirect effects. Moreover, we respectfully suggest a new experimental program is beyond the scope of this manuscript.

Line 472: “Notably, the extension of the SO-1 binding surface onto β-sheet 3 and the loop results in large overlap with the binding surface involved in the Cyp33-RRM and MLL1-PHD3 interaction” – are the magnitudes of chemical shifts significant? The reviewer is asking about the magnitude of the chemical shift changes upon SO-1 binding, and as can now be seen in Figure 4c, indeed the magnitude of chemical shift change is similar in the canonical and non-cannonical regions. However, the magnitude of chemical shift change does not directly reflect the extent of structural engagement at that site. We believe it is difficult to reliably interpret magnitudes of chemical shift changes. As seen from the histogram and examination of the titration data, it is clear the changes are significant (i.e., not noise) and represent a change in the chemical environment at that site.

Line 485 and Supplementary Figure 7: The message here that extra regions of the FL Cyp33 protein is binding to the SO-1 is not discernible from this Figure. As the spectrum has not yet been fully assigned, concluding drawing form conclusions about expanded binding regions for SO-1 beyond the RRM domain is premature in the absence of addition data such as effects of mutagenesis. The fact that the linewidths of the spectra are similar means that binding is very weak indeed. Two points here to address – the first is that we observe (Supp Figure 4) that the free full-length Cyp33 spectrum is well-represented by the sum of the Cyp33-RRM and Cyp33-Cyp spectra. Consistent with this, in Supp Figure 7, we observe the Cyp33-RRM shifts upon SO-1 binding recapitulated in the full-length protein (now labeled to make the point more clearly). The assigned Cyp33-RRM peaks that do not shift upon binding are likewise unpertubed in this spectrum. We observe additional shifts outside of the ones ascribed to the Cyp33-RRM (unassigned, now circled in pink so they can be readily detected), which we attribute to elements outside of the RRM domain. In light of the reviewer’s concern, we now present a more conservative interpretation and have eliminated the speculation that these lie in the linker, just note that they lie outside of the RRM. We note that this NMR data provides structural support for the biochemical observation that full-length Cyp33 binds SO-1 30-fold more tightly than the isolated RRM domain, suggesting an additional surface is present in the full-length protein and its activity is not fully recapitulated by the RRM domain.

 The primary binding event of SO-1 to the RRM is in the fast exchange regime, consistent with the binding affinity measured. As noted below, upon re-examination of our data in light of the reviewers comments we are no longer commenting on the exchange regime as it is difficult to see unambiguously in the data. This does not change any of our conclusions.

Line 495: “However, several chemical shifts not attributable to the known domains exhibit slow-exchange peak shifting behaviour. As these cannot be ascribed to either the Cyp33-RRM or Cyp33-Cyp domains, it suggests the involvement of the 46 amino acid linker in the interaction peaks”. How many peaks are in slow exchange? We have highlighted the 7 clearest examples of peaks shifting outside of the assigned RRM resonances, circled now in pink in Supp Figure 7. 

Do the binding site residues in the Cyp33 RRM domain also display slow exchange? If not, why are the linker residues in slow exchange? Slow exchange can be a result of sample conditions – was an excess of SO-1 added the Cyp33Fl sample to try and shift the equilibrium more in favour of the bound form? We appreciate the reviewer’s concerns with the interpretation of these data and have qualified some of our conclusions to reflect a more conservative interpretation of the chemical shift data, as described in the response above and in the text at lines 503-511. Specifically, we have removed discussion of the “slow exchange” regime and tempered our conclusion of the involvement of the linker. As the reviewer correctly points out, it is an overreach to make conclusions in the absence of independent assignments.

Line502, Figure 5: The shift changes should be displayed as histograms for the results to be interpretable. The data mapped on this figure were obtained from Hom et al. (reference #7) and they unfortunately did not provide more quantitative data for interpretation. We did not repeat the published experiment, we instead started with an RNA-bound sample and titrated in the peptide.

Line515: what are the relative binding constants between the Cyp33 RRM domain to SO-1 and to PHD3. Would you not expect the RRM domain to have more preference for the RNA than the PHD3? Was the reverse NMR competition experiment performed using an excess of PHD3 to compete out the RRM domain? Thank you for this question, as we believe addressing it more careful strengthens our paper. We measured the Kd for the Cyp33 RRM to SO-1 using gel shift, and report that Kd along with the representative gel in Supplementary Figure 3. We find the Kd for RNA is weak, at 180 uM. This is fully consistent with published data, where the RRM bound a short RNA (AAUAAA) with 200 uM affinity, and in comparison to the Kd for the peptide, reported by Hom to be 1.9 uM. We allude to this binding preference now in the text (Lines 528-530). We did not have the material to perform the titration in reverse, but we do not believe that is necessary to show that the interaction is competitive.

---

## [Editor Report · Decision Letter 1]

26 Jan 2021

Cyp33 Binds AU-Rich RNA Motifs via an Extended Interface that Competitively Disrupts the Gene Repressive Cyp33-MLL1 Interaction in vitro

PONE-D-20-24207R1

Dear Dr. Wuttke,

We’re pleased to inform you that your manuscript has been judged scientifically suitable for publication and will be formally accepted for publication once it meets all outstanding technical requirements.

Kind regards,

Oscar Millet

Academic Editor

PLOS ONE
---

## [Editor Report · Acceptance letter]

29 Jan 2021

PONE-D-20-24207R1 

Cyp33 Binds AU-Rich RNA Motifs via an Extended Interface that Competitively Disrupts the Gene Repressive Cyp33-MLL1 Interaction *in vitro*

Dear Dr. Wuttke:

I'm pleased to inform you that your manuscript has been deemed suitable for publication in PLOS ONE. Congratulations! Your manuscript is now with our production department. 

Kind regards, 

on behalf of

Dr. Oscar Millet 

Academic Editor

PLOS ONE